# RAD51D Secondary Mutation-Mediated Resistance to PARP-Inhibitor-Based Therapy in HGSOC

**DOI:** 10.3390/ijms241914476

**Published:** 2023-09-23

**Authors:** Jing Xu, Yilin Dai, Yi Gao, Ranran Chai, Chong Lu, Bing Yu, Yu Kang, Congjian Xu

**Affiliations:** 1Obstetrics and Gynecology Hospital of Fudan University, Shanghai 200011, Chinaxucongjian@fudan.edu.cn (C.X.); 2Shanghai Key Laboratory of Female Reproductive Endocrine Related Diseases, Shanghai 200011, China; 3Central Clinical School, The University of Sydney, Sydney, NSW 2006, Australia

**Keywords:** RAD51D, ovarian cancer, poly(ADP-ribose) polymerase inhibitors, chemo-resistance

## Abstract

Ovarian cancer is the leading cause of gynecologic cancer-related death, and PARP inhibitors (PARPis) are becoming a promising treatment option, as demonstrated by recent clinical trials. After PARPi exposure, somatic reversion mutations in the homologous recombination genes may be a mechanism of PARPi resistance in ovarian carcinoma. We present an ovarian cancer case of a 61-year-old woman, who underwent routine tumor reduction surgery followed by platinum and PARPis. She demonstrated a good response to PARPis for 15 months before recurrence and secondary tumor reduction surgery. However, post-surgery platinum and PARPi treatment only kept the disease stable for 5 months. A potential molecular mechanism for PARPi resistance was investigated using next-generation sequencing, immunohistochemical (IHC) staining, and other functional assays. A germline RAD51D loss-of-function mutation was found in the reported case (LRG_516t1:c.270_271dup p1:p.(Lys91fs*13)). Subsequently, a secondary mutation (LRG_516t1:c.271_282 del) was identified in the same locus of the germline duplication in the post-progression biopsies and ctDNA. The IHC staining supported low expression of RAD51D in the initial tumor tissue, but the expression was restored after the correction of the open reading frame by the secondary mutation. The in vitro results supported that the loss-of-function mutation of RAD51D was the basis for the initial response to the platinum and PARPi therapy, while the newly acquired reversion mutation could be attributed to the observed PARPi resistance. An acquired mutation can reverse a loss-of-function change in RAD51D and can result in PARPi resistance in a hereditary ovarian cancer patient. Liquid biopsy could be considered for longitudinal monitoring in ovarian patients under PARPi-based therapy, which can identify acquired resistant mutations earlier and facilitate precision management.

## 1. Introduction

Ovarian cancer is an aggressive gynecologic malignancy [1], and the approval of poly ADP ribose polymerase inhibitors (PARPis) since 2018 highlights the utility of synthetic lethality as a therapeutic approach in the treatment of ovarian cancer [2]. Germline or somatic mutations in homologous recombination (HR) genes have been shown to respond well to PARPis in vitro and in patients [3]. Cells lacking HR repair must deal with double-strand DNA breaks through more error-prone forms of DNA repair such as non-homologous end joining [4]. Therefore, HR-deficient ovarian tumors show an increased sensitivity to PARP1 inhibitors, given that PARPis induce double-strand breaks either directly or through the stalling and subsequent collapse of replication forks [5].

Although long-term responders to PARPis exist, most cancer patients ultimately become resistant to PARPi therapy. Somatic reversion mutations in HR genes following exposure of ovarian carcinoma to PARPis have been reported as mechanisms of such resistance [6]. Reversion mutation refers to an acquired change that can restore or partially restore the protein expression and functional capacity that is lost due to a primary loss-of-function variant. Multiple mechanisms have been proposed, for example, acquired in-frame deletions removing the region of primary frameshift mutations or splice-site mutations [7].

BRCA1/2’s reversion mutations were first reported in 2008, but the studies have largely been related to platinum resistance, with limited reports on tumors from patients treated with PARPis [8]. Apart from BRCA1/2, recent studies have revealed that the somatic reversion mutations of the core HR genes, RAD51C and RAD51D, might be mechanisms of acquired resistance to the PARPis, rucaparib, in ovarian cancers [9].

Here, we describe a case of a patient who developed resistance to PARPi therapy (Niraparib, Olaparib) because of a reversion mutation of RAD51D at the same locus of the germline mutation.

## 2. Results

### 2.1. Case Presentation

A sixty-one-year-old Chinese woman was diagnosed with HGSOC and received her first tumor reduction surgery in April 2015, followed by paclitaxel plus platinum only for three courses due to severe myelosuppression and intestinal obstruction (Figure 1). In January 2017, an MRI of the pelvic cavity showed multiple nodules in the liver, spleen, and kidneys, especially a 4 cm mass at the anastomosis of the abdominal wall, which indicated the possibility of metastasis. So, the patient was given the oral PARPi niraparib for 11 months, and then switched to oral olaparib for 2 months. In the meantime, the CA125 value fluctuated between 67 and 576 U/mL. In January 2018, the PET-CT revealed multiple metastases. The patient was referred to the Obstetrics and Gynecology Hospital of Fudan University and underwent a second tumor resection surgery in April 2018. However, the platinum and PARPi after the secondary surgery only kept the disease stable for 5 months. The MRI revealed metastatic nodules in the abdominal cavity and left lower abdomen, which means that the patient developed resistance (Figure 1).

### 2.2. Molecular Study

A germline 2bp-duplication was identified (LRG_516t1:c.270_271dup), which resulted in a frameshift (LRG_516p1:p.(Lys91fs*13)) and predicted the null expression of RAD51D due to nonsense-mediated mRNA decay (Figure 2A). This germline mutation was actually passed down to the next generation (Figure 2A), and we need to pay attention to the screening and genetic counseling of the next generation. A secondary mutation was identified (LRG_516t1:c.271_282del) from the tumor sample of the secondary surgery and the liver biopsy at the post-progression stage after a chemotherapy regime with docetaxel and cyclophosphamide. This acquired 12 bp deletion was overlapped with the germline mutation, eliminated the frame-shift effect of the 2 bp duplication, and restored RAD51D expression (LRG_516p1:p.(Lys91_Asp94del), Figure 2B). Blood was also taken for ctDNA isolation at the secondary surgery and the post-progression stage, and the secondary mutation with VAF 0.7% was also found in the liquid biopsy analysis. As a result, the restored RAD51D was assumed to resume the HR function and thus confer resistance to PARPi, and the reverse mutation could be detected via longitudinal monitoring from ctDNA.

### 2.3. Immunohistochemical Analysis

Figure 3 demonstrates different RAD51D expression in the tumor samples derived from the first and second surgeries, as well as from the recurrent tumor at the PARPi-resistant stage. As indicated in the molecular study, the germline duplication resulted in a frameshift change and led to loss-of-function for RAD51D. This is supported by the immunohistochemical (IHC) results (Figure 3A). Low IHC staining of RAD51D was observed in the tumor section derived from the first surgery, which resulted in the HR defect and it becoming sensitive to PARPi.

As suggested by the molecular study, the acquiring of the 12 bp deletion is correlated with resistance to the PARP-inhibitor-based therapy. This secondary mutation can overwrite the inherited frameshift duplication and resume the reading frame and RAD51D expression, as shown in the IHC staining of the samples from the second surgery and liver biopsy (Figure 3A) [9].

### 2.4. Functional Investigation

RAD51 and γ-H2AX foci formation assays illustrated that deficient HR repair in the first surgery tumor samples was complemented by the primary RAD51D mutation but not with the secondary RAD51D mutation (Figure 3B).

The functional implication of the primary and secondary mutations was investigated in vitro using OVCAR8 and SKOV3 (ovarian cancer cell lines) and HELA (a cervical adenocarcinoma cell line). Western blotting was used to confirm the overexpression status of the expected clones (Figure 4).

The γ-H2AX foci formation assays confirmed proficient HR repair in cells complemented with wildtype RAD51D or any of the secondary RAD51D mutations tested, but not with the primary RAD51D mutation (Figure 5).

By assessing cell viability subsequent to treatment with multiple other PARP inhibitors (olaparib and niraparib) at specified concentrations, our findings indicated that introduction of RAD51D cDNA with the primary mutation (to the KO cells) led to an improved sensitivity to PARPi. This effect could be reverted and entered the resistant stage to PARPi if wildtype RAD51D or RAD51D with the secondary mutation was introduced (Figure 6).

Such in vitro results further support the assertion that the new reversion mutation can be attributed to PARPi resistance in this patient.

## 3. Discussion

Currently, the mechanisms of PARP inhibitor resistance can be generally classified into the following categories [11]: (1) cellular availability of the inhibitor; (2) the activity and abundance of PAR chains; (3) reactivation of the homologous recombination; and (4) replication fork protection. The restoration of HR can occur due to secondary mutations that remove or compensate for the original genetic lesion. Among the resistant mechanisms of PARPis, a secondary mutation is relatively common. It has been reported since 2008 [8], and was initially found in the second mutation of BRCA. Later, studies also revealed secondary mutations in RAD51C and RAD51D as a mechanism of acquired PARPi resistance.

This case report illustrated again that the patient’s clinical benefit was associated with RAD51D frameshift mutation leading to a loss of RAD51D expression. Our in vitro assay reproduced the PARPi sensitivity of the inherited 2 bp duplication, and RAD51D and γ-H2AX foci formation assays confirmed that deficient HR repair existed in the first surgery tumor sample (Figure 3B). This is the molecular basis for why the patient had a partial response to niraparib based on RECIST (Response Evaluation in Solid Tumor) and enjoyed 12 months of progression-free survival.

We reported a new reversion mutation with RAD51D 12 bp deletion that conferred resistance to PARPi-based therapy in this ovarian cancer patient. Its functional impact was substantiated by RAD51 and γ-H2AX foci formation assays, which illustrated that proficient HR repair was complemented by the secondary RAD51D mutation (Figure 3). The difference in RAD51D protein expression in tumor samples between the first surgery, second surgery, and liver biopsy demonstrated the roles of primary mutation and secondary mutation in RAD51D function. In addition, in the first surgery tumor sample, we found that although the RAD51D protein was basically not expressed, there were still some scattered expressed lesions, which might be PARPi-resistant and might be expanded under the selection pressure imposed by PARPi exposure, ultimately leading to drug resistance.

Next-generation sequencing (NGS) was conducted for the RAD51D genes in a 781 Chinese ovarian cancer patient cohort by the Chinese Academy of Medical Sciences and Peking Union Medical College, Beijing. The results showed that RAD51D germline pathogenic mutations were detected in 1.7% (13/781) of patients and the RAD51D c. 270_271dup (p. Lys91fs) mutation was the most common mutation, found in seven patients (7/13, 53.1%) [12]. Given the high frequency of this mutation site in Chinese patients, we hypothesize that it may be the founder mutation in the Chinese population.

In response to the increasing number of cases of PARPi resistance, future studies are needed to understand the mechanism and how it may impact the choice of combination strategies in PARPi-resistant ovarian cancer patients. For this patient, based on previous research and drug-sensitivity data, which showed that AKTi and albumin paclitaxel may combat this resistance, our patient did have a good response to the clinical use of albumin paclitaxel and reached PR status.

It is also important to note that this resistance mutation could be detected in the liquid biopsy. Ovarian cancer patients under PARPi treatment should be monitored beyond conventional follow up with imaging and CA125. Liquid biopsy can identify acquired resistant mutations earlier and facilitate the change in treatment strategy to delay or overcome a potential resistance.

## 4. Materials and Methods

### 4.1. Tumor Tissue and ctDNA

This study was conducted in accordance with the Declaration of Helsinki and approved by the ethical committee of the Obstetrics and Gynecology Hospital of Fudan University (Number: 2021-174).

All of the blood and tumor samples were collected from a 61-year-old Chinese woman with high-grade serous ovarian cancer (HGSOC). DNA extracted from peripheral white blood cells was used for the germline mutation analysis. Tumor samples were derived from primary and secondary surgeries and a biopsy from a metastatic liver. Such tissue samples were fixed in 10% neutral formalin for 24 h to generate FFPE block.

Then, 10 mL of blood was collected from this patient for next-generation sequencing (NGS) of circulating tumor DNA (ctDNA) and identification of germline mutation and somatic mutation. For plasma collected using Streck BCT, cfDNA was isolated using the Qiagen QIAamp Circulating Nucleic Acid Kit (Qiagen, Duesseldorf, Germany, Cat# 55114), according to the manufacturer’s protocol. Repeated freezing and thawing of plasma were avoided to prevent cfDNA degradation and gDNA contamination from white blood cells (WBC). The concentration of cfDNA was measured using the Qubit dsDNA HS Assay Kit (Thermo Fisher Scientific, Waltham, MA, USA, Cat# Q32854), and the quality was examined using the Agilent High Sensitivity DNA Kit (Cat# 5067-4626). cfDNA with yields greater than 5 ng without overly genomic DNA contamination proceeded to library construction. Library construction was performed using the KAPA HyperPlus Kit (Roche, Basel, Switzerland, Cat# KK8504), and target enrichment was performed using Target Probes IGT Kit (iGene Tech, Beijing, China, cat# T232 V2).

### 4.2. NGS Sequencing

Next-generation sequencing (NGS) was used to identify germline and somatic mutations. At least 30 ng of genomic DNA was isolated from samples and subsequently sheared using the Covaris E220 instrument (Covaris, Woburn, MA, USA). Sequence libraries were prepared using the KAPA HyperPlus Library Preparation Kit by first producing blunt ends, and 5-phosphorylated fragment strand dAMP was added (A-tailing) to the 3′ ends of the dsDNA library fragments. Next, dsDNA adapters with 3′-dTMP were ligated to the A-tailed library fragments. Library fragments with the appropriate adapter sequences were amplified via ligation-mediated pre-capture PCR. Library capture was conducted using a custom probe system (IGene Tech, Beijing, China; Covaris, Woburn, MA, USA) and biotinylated to allow sequence enrichment by capture using streptavidin conjugated beads (Thermo Fisher, Waltham, MA, USA). Eventually, the pooled libraries containing captured DNA fragments were sequenced using the NextSeq 500/550 High Output Kit v2.5 (300 Cycles on the Illumina NextSeq 500 system) to produce 150 bp paired-end reads.

### 4.3. Immunofluorescent and Immunohistochemical Analyses

Antigen retrieval was performed on the FFPE sections using citrate buffer (0.1 mol/L, pH 6.0) with a steamer. Serial 4 µm sections were incubated with primary antibodies (RAD51, sc-398587,1:250, Santa Cruz; γ-H2AX, ab22551,1:200, Abcam) at 4 °C overnight. Subsequently, the samples were probed with a secondary antibody. Finally, the samples were mounted with DAPI-containing mounting medium (S36973, Thermo Fisher, Waltham, MA, USA). Images were captured with a fluorescence microscope (Leica, Wetzlar, Germany) with Leica Application Suite V4 software and edited with Photoshop (Adobe, USA).

Serial 4 µm sections were incubated with primary antibodies (RAD51D, ab202063, 1:250, Abcam, Cambridge, MA, USA) overnight at 4 °C. Sections without primary antibody incubation were used as negative controls. Goat anti-mouse horseradish peroxidase–conjugated secondary antibody was used. The chromogenic reaction was performed with DAB.

### 4.4. Plasmids and Lentiviral Production

The following single guide (sg) RNA primers were used to knock out RAD51D: forward: 5′-CACCGAAGCAGTTTATCAAGACTGA-3′, and reverse: 5′-AAACTCAGTCTTGATAAACTGCTTC-3′.

The sgRNA sequences were annealed and inserted into the PX458-vector (Addgene, Plasmid #48138).

Different plasmids were constructed to overexpress wildtype and mutant RAD51D. The amplified RAD51D coding sequences with wildtype or with the primary (LRG_516t1: c.270_271dup) or secondary (LRG_516t1: c.271_282del) mutation were inserted into the pGMLV lentiviral vectors (Genomeditech Co., Ltd., Shanghai, China).

Human embryonic kidney (HEK-293T) cells were transfected with the lentiviral expression vectors and packaging plasmids for lentivirus production. Lentiviral supernatants were collected for 48 h after transfection, filtered, and concentrated via centrifugation.

### 4.5. Generation of RAD51D KO and Overexpressing Cell Lines

The cell lines utilized in this study were purchased from the American Type Culture Collection (Manassas, VA, USA).

To generate RAD51D KO cells, HELA, OVCAR8, and SKOV3 cells were electroporated with PX458. After trypsinization and centrifugation (10 min, 90× *g*), the cells were counted, and 1 × 10^6^ of each sample of cells were resuspended in 100 μL of 4D-Nucleofector Solution with 10 μg RAD51D CRISPR plasmid. The master mixes were transferred into the Nucleocuvette Vessels that were placed into the retainer of the 4D-Nucleofector X Unit. Then, the corresponding nucleofection program (FE-132 for OVCAR8 and SKOV3, CN-114 for HELA) was initiated. After electroporation, the cells were resuspended with pre-warmed culture medium and added into pre-warmed 6-well cell culture plates. The medium was replaced the following day. On day 3, GFP-positive cells were single-cell plated using flow cytometry (BD FACSAria Fusion, BD Bioscience, San Jose, CA, USA).

After cell expansion for at least 2 weeks, RAD51D KO was confirmed in the single-cell colonies by DNA sequencing and Western blotting. To generate cells that stably express wildtype and mutated RAD51D, these RAD51D KO single-cell colonies were transduced with the lentiviruses mentioned above. The transduced cells were selected by continuous puromycin treatment (MedChem Express, Monmouth Junction, NJ, USA). Western blotting was used to confirm the overexpression status of the expected clones. The following primary antibodies were used: RAD51D, sc-398819,1:500, Santa Cruz Biotechnology, Santa Cruz, CA, USA; Flag, F3165,1:1000, MilliporeSigma, Billerica, MA, USA; GAPDH, 60004-1-Ig, 1:7500, Proteintech, Wuhan, China; β-actin, sc-47778,1:5000, Santa Cruz Biotechnology, Santa Cruz, CA, USA.

### 4.6. γH2AX Foci Formation Assay

HELA cells were treated with dimethyl sulfoxide (DMSO) or DMSO with 10 μmol/L niraparib (Selleck, Houston, TX, USA) for 24 h. The cells were fixed with 4% paraformaldehyde, permeabilized with PBT (0.25% TritonX-100 in PBS), blocked with blocking buffer (5% BSA in PBT), and incubated with rabbit anti-γH2AX (1:200, 9718, Cell Signaling Technology, Danvers, MA, USA). Anti-rabbit Cy3 was used as a secondary antibody. Nuclei were counterstained with DAPI (S36973, Thermo Fisher, Waltham, MA, USA). The cells were imaged at 60× using a Confocal Laser Scanning Microscope FV3000 (Olympus GmbH, Hamburg, Germany). At least 50 cells per cell line were counted using FociCounter [10].

### 4.7. Cell Viability Assays

Cell viability was measured using a commercial CCK-8 assay kit (APExBIO, Houston, TX, USA). The cells were plated in 96-well plates at a density of 2000 per well and grown for 24 h. Then, the cells were treated with fresh media containing various concentrations of olaparib, niraparib, and rucaparib (Selleck, Houston, TX, USA) for 72 h. Each well was treated with 10 μL of CCK-8 reagent; then, the cells were incubated at 37 °C for another 4 h before measuring the absorbance at 450 nm using microplate reader (Perkin Elmer, Waltham, MA, USA).

### 4.8. Statistical Analysis

All of the assays were performed at least in triplicate. All experimental data were analyzed using GraphPad Prism v. 8.0 software (GraphPad, San Diego, CA, USA). One-way analysis of variance (ANOVA) followed by the Tukey test was used to compare the means of band intensity values of the Western blot and γH2AX foci formation assays. Data were presented as the mean ± standard error of the mean (SEM). Half-maximal inhibitory concentration (IC50) values were calculated from nonlinear regression curves (dose–response curves). * *p* < 0.05, ** *p* < 0.01, *** *p* < 0.001, and ns (no significant) were considered as the statistical significance.

## 5. Conclusions

In summary, our data supported the assertion that secondary mutation in RAD51D can not only reverse a loss-of-function change of RAD51D that was disrupted by the inherited defect in an ovarian cancer patient but also reinstitute HR function and contribute to the development of clinical resistance to PARPi. The liquid biopsy result is also proof-of-principle that the reverse mutation can be detected via longitudinal monitoring from ctDNA and can identify acquired resistant mutations earlier.

## Figures and Tables

**Figure 1 ijms-24-14476-f001:**
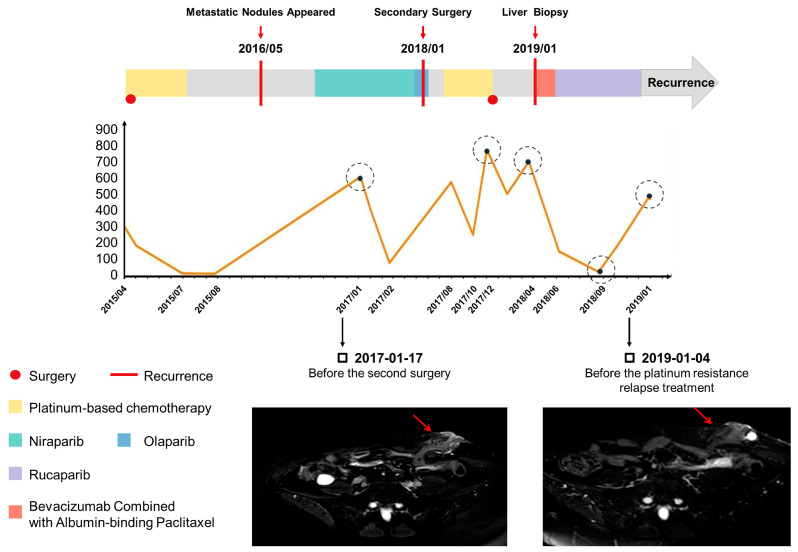
Patient‘s clinical presentation. The graphic shows the disease course and clinical findings of the patient, including CA125 levels and MRI images over time. The value of the CA125 level reflects the patient’s response to treatment during the disease course, indicating when the patient may develop drug resistance. The MRI in 2017 showed metastatic lesions prior to treatment with PARPis. The MRI in 2019 showed another recurrence after the second surgery with a TC regimen combined with PARPi maintenance therapy. The mass marked in two MRI images represents the metastatic nodules of anastomosis of the abdominal wall during the patient’s first surgery.

**Figure 2 ijms-24-14476-f002:**
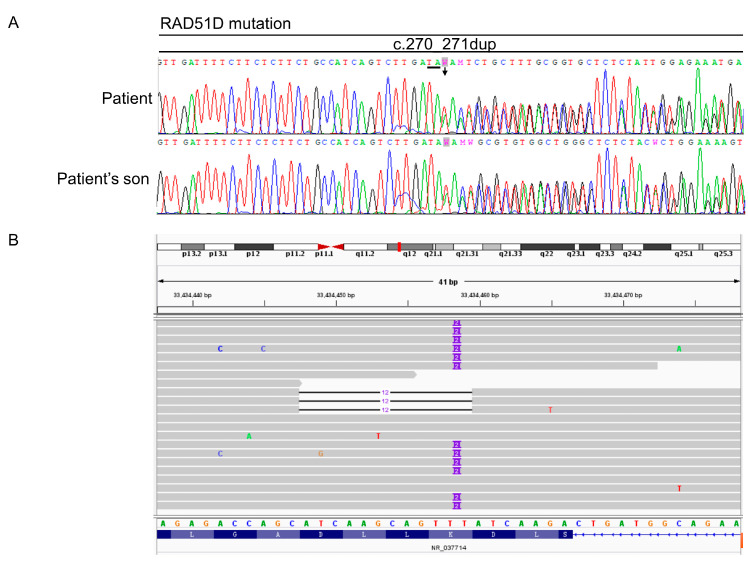
Primary mutation and secondary mutation of RAD51D. (**A**) Samples from the patient and her son with germline RAD51D mutation. (**B**) Diagram of the RAD51D reversion mutation alleles observed in the patient.

**Figure 3 ijms-24-14476-f003:**
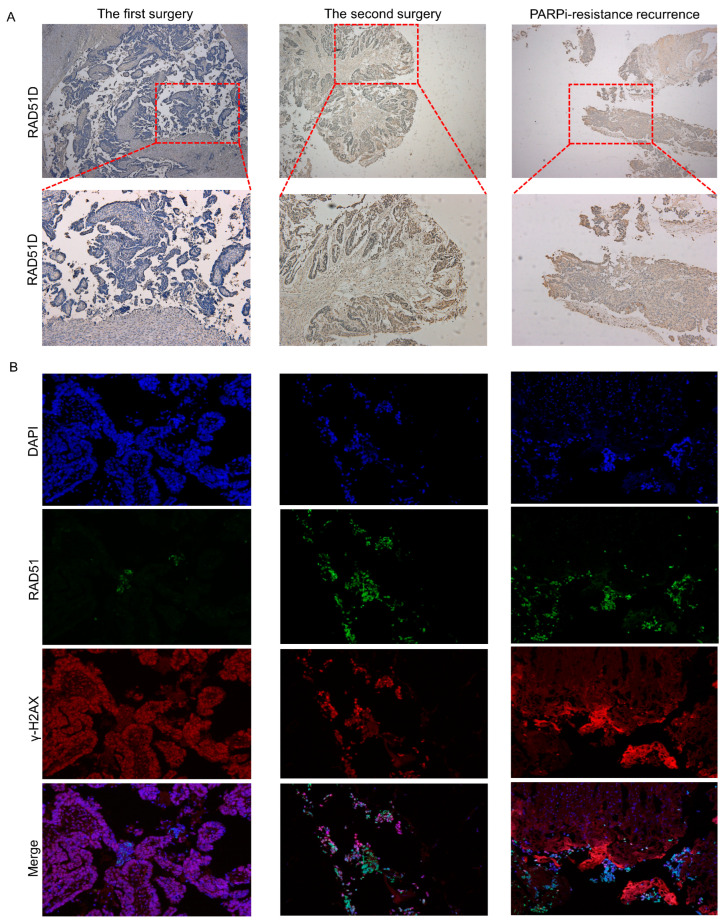
Expression of RAD51D ((**A**) 40× in the upper line and 100× in the lower line) and HR function (**B**) in tumor samples from the first surgery, second surgery, and liver biopsy. No expression of RAD51D in the first surgery tumor samples supported the assertion that the germline frameshift can lead to non-sense mediated decay (**A**) and resulted in a loss of HR function (**B**). RAD51D expression restored in the secondary surgery samples including PARPi resistant tissue from liver biopsies. This demonstrates that this secondary somatic deletion could restore the reading frame of the RAD51D gene and resume RAD51D function (**A**) and, consequently, HR repair function (**B**).

**Figure 4 ijms-24-14476-f004:**
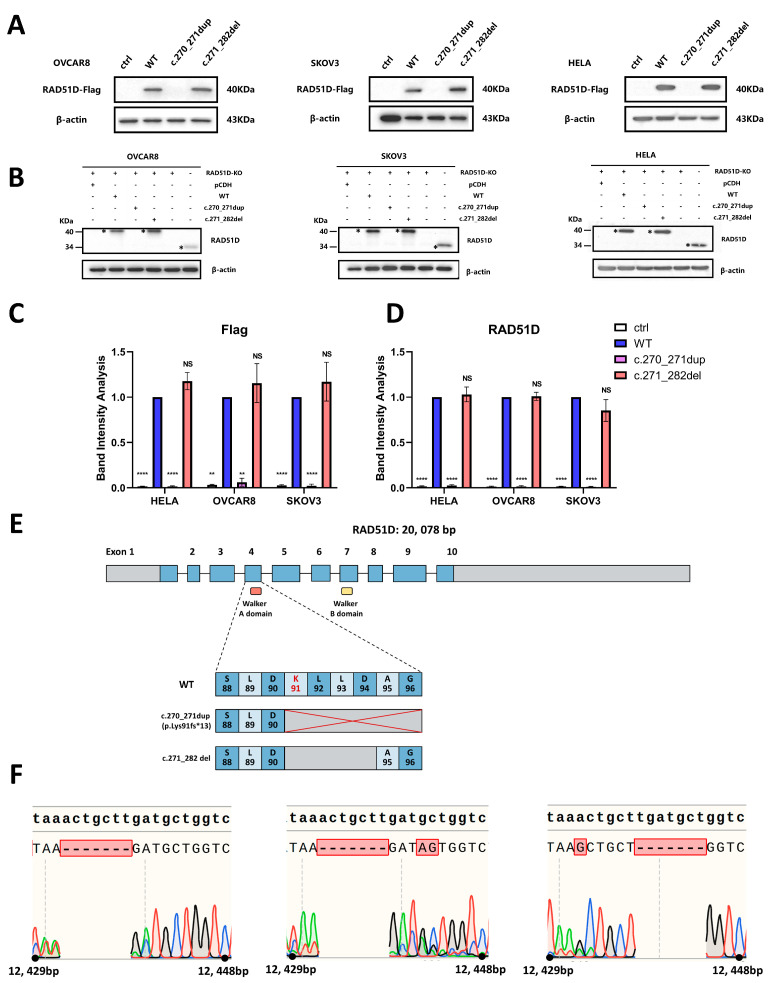
Verification of the RAD51D-KO clones of the OVCAR8, SKOV3, and HELA cell lines transduced with WT or primary or secondary mutant RAD51D transcripts. (**A**,**B**) Western blotting of Flag (**A**), RAD51D (**B**), and β-actin (control) protein expression in RAD51D-KO clones of OVCAR8, SKOV3, and HELA cell lines transduced with WT, primary or secondary mutant RAD51D transcripts. (**C**,**D**) Statistical analysis of the band intensity values of Flag (**C**) and RAD51D (**D**). (**E**) A diagram of alternations in the RAD51D protein sequence resulting from the primary mutation (LRG_516t1:c.270_271dup) and the secondary mutation (LRG_516t1:c.271_282del) identified in the patient. (**F**) Genomic DNA sanger sequencing of RAD51D in parental OVCAR8, SKOV3, and HELA cell lines and the corresponding RAD51D KO clones. Color code of the sanger sequencing: red = T, thymine; green = A, adenine; blue = C, cytosine; black = G. guanine. NS = no significant, *p* > 0.05; **, *p* < 0.01; ****, *p* < 0.0001.

**Figure 5 ijms-24-14476-f005:**
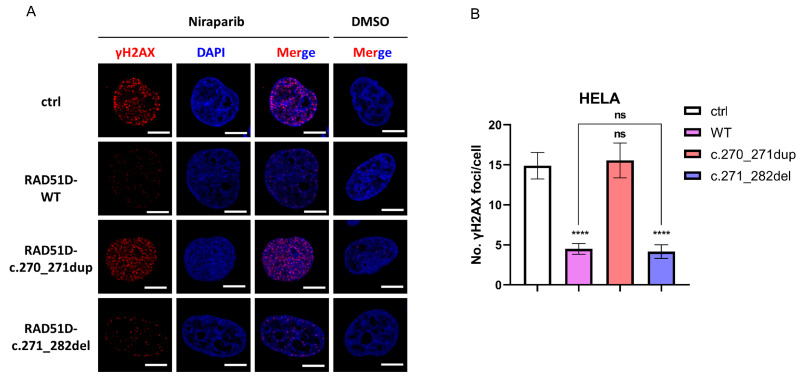
HELA cells transduced with empty vector RAD51D-WT, RAD51D-c.270_271dup, or RAD51D-c.271_282del. (**A**) Immunofluorescence staining with γH2AX antibody after treatment with niraparib (10 μmol for 24 h) or dimethyl sulfoxide (DMSO). Cell nuclei were counterstained with DAPI (scale bar = 10 μm). (**B**) Quantitative results of the γH2AX foci. Overexpression of RAD51D WT or RAD51D with the secondary mutation led to a statistically significant reduction in γH2AX foci. In contrast, no statistically significant decrease in γH2AX foci was observed in cells overexpressing RAD51D with the primary mutation. At least 50 cells per cell line were counted using FociCounter [10]. NS = no significant, *p* > 0.05; ****, *p* < 0.0001.

**Figure 6 ijms-24-14476-f006:**
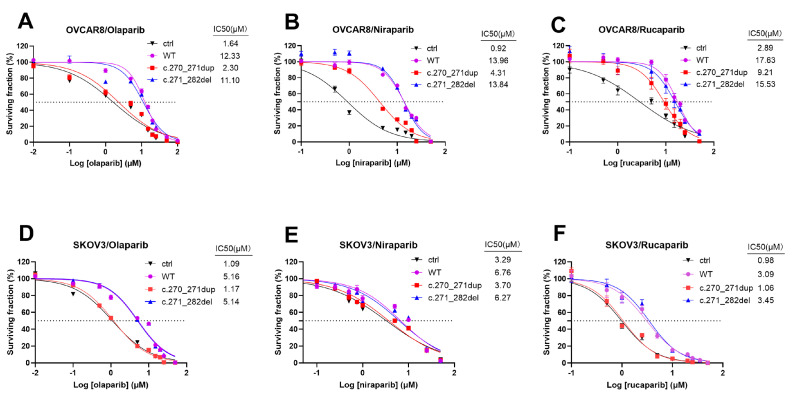
Cell viability following treatment with olaparib, niraparib, or rucaparib at the indicated concentration for a duration of 72 h measured in OVCAR8 (**A**−**C**) and SKOV3 (**D**−**F**) cells, which stably transduced with empty vector, RAD51D−WT, RAD51D−c.270_271dup, or RAD51D−c.271_282del.

## Data Availability

Data sharing for patient-specific NGS sequencing results is not applicable.

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
