# Peer review of "RAD51D Secondary Mutation-Mediated Resistance to PARP-Inhibitor-Based Therapy in HGSOC"

_ijms, 2023, doi:10.3390/ijms241914476_

Round 1

Reviewer 1 Report

The researchers focused on ovarian cancer, a leading cause of gynecologic cancer death. They studied PARP inhibitors (PARPi) as a hopeful treatment, backed by recent trials. They explored PARPi resistance, which might arise from gene mutations after PARPi exposure.

They presented a case of a 61-year-old ovarian cancer patient. After routine tumor reduction surgery, she received platinum and PARPi, resulting in a positive 15-month response. Recurrence led to another surgery and subsequent platinum/PARPi treatment, stabilizing the disease for just 5 months.

All experiments were meticulously planned and executed using advanced sequencing, immunohistochemistry, and functional assays. A primary RAD51D gene mutation (loss-of-function) was identified, followed by a secondary mutation post-progression and in ctDNA. Initial low RAD51D expression was restored by the secondary mutation.

In vitro experiments validated that the initial response resulted from the loss-of-function mutation, while the acquired reversion mutation correlated with PARPi resistance.

In conclusion, they demonstrated that acquired mutations can reverse loss-of-function changes, causing PARPi resistance in hereditary ovarian cancer. They suggested liquid biopsies for ongoing monitoring during PARPi-based therapy to identify resistant mutations early for better management.

Author Response

Dear reviewer,

Thank you for reviewing our manuscript and providing these constructive comments. We have gone through them one by one and made revisions in the manuscript accordingly. Please see the attachment. Following is our response (in red font) to each of your comment. We have also specified the revision we made in the response. Please review and let us know your thoughts.

Best regards!

Yours sincerely,

Yilin Dai

Reviewer #1:

The researchers focused on ovarian cancer, a leading cause of gynecologic cancer death. They studied PARP inhibitors (PARPi) as a hopeful treatment, backed by recent trials. They explored PARPi resistance, which might arise from gene mutations after PARPi exposure.

They presented a case of a 61-year-old ovarian cancer patient. After routine tumor reduction surgery, she received platinum and PARPi, resulting in a positive 15-month response. Recurrence led to another surgery and subsequent platinum/PARPi treatment, stabilizing the disease for just 5 months.

All experiments were meticulously planned and executed using advanced sequencing, immunohistochemistry, and functional assays. A primary RAD51D gene mutation (loss-of-function) was identified, followed by a secondary mutation post-progression and in ctDNA. Initial low RAD51D expression was restored by the secondary mutation.

In vitro experiments validated that the initial response resulted from the loss-of-function mutation, while the acquired reversion mutation correlated with PARPi resistance.

In conclusion, they demonstrated that acquired mutations can reverse loss-of-function changes, causing PARPi resistance in hereditary ovarian cancer. They suggested liquid biopsies for ongoing monitoring during PARPi-based therapy to identify resistant mutations early for better management.

Response: Thank you for reviewing our manuscript and acknowledging our conclusions.

Reviewer 2 Report

In this article, the authors present patient data (n=1) with corroborating in-vitro data implicating the rise in RAD51D secondary mutation to the rise in therapeutic resistance to PARPi. Similar studies documenting or stating presence of a deletion mutation in RAD51D and its role in PAPRi resistance in high grade ovarian cancer were published in earlier studies (doi.org/10.1158/2159-8290.CD-17-0419; doi.org/10.1016/j.tranon.2021.101012; doi.org/10.1111/jcmm.14133; doi.org/10.1158/0008-5472.CAN-17-0190; doi.org/10.1038/s41467-021-22582-6). While the data expands on the causative role, specific evidence linking the gene to the rise in molecular resistance may still be required. The motivation in the article in clear and the data is within the scope of the journal. Some additional data to support the mechanistic investigation into the molecular biology of resistance generation may be required (such as studying expression of downstream/upstream protein and associated pathways of the gene in question) In addition, the manuscript has a lot of typographical errors such as inclusion of unnecessary dashes in words, spelling issues etc. I would request the authors to meticulously check for all errors for corrections. Several of the scientific issues can be corrected based on the comments/recommendations below:

1.       (line 114) “secondary mutation can overwrite the frameshift duplication as inherited” – please add relevant citations.

2.       In Figure 2, is there a comparison sequence from patient prior to therapy/developing resistance?

3.       Have the authors conducted invasion/migration study and expression of EMT markers to assess role of the secondary mutation and resistance? Such data may be helpful.

4.       The authors can mention the duration of treatment in the captions for Figure 5.

5.       Supplementary figure-1 should be included within the manuscript.

6.       Have the authors conducted western blot triplicates? If yes, the significance between the band intensity values should be plotted with relevant p-values.

7.       Please add a separate figure in supplementary data showing the full-length sequences with x-axis for Figure S1-c.

8.       Have the authors conducted any rtPCR analysis for abnormal gene amplification in RAD51 (wt and mutation) expressing cells? Such data may be pertinent to understand the mechanism behind resistance generation.

9.       Please add statistical analysis section in methods. All appropriate methods/tests and software used should be noted.

10.   Please list the type/model of flow cytometry used.

11.   Authors need to mention where cell lines were obtained from in the methods section.

The manuscript has a lot of typographical errors such as inclusion of unnecessary dashes in words, spelling issues etc. I would request the authors to meticulously check for all errors for corrections.

Author Response

Dear reviewer,

Thank you for reviewing our manuscript and providing these constructive comments. We have gone through them one by one and made revisions in the manuscript accordingly. Please see the attachment. Following is our response (in red font) to each of your comment. We have also specified the revision we made in the response. Please review and let us know your thoughts.

Best regards!

Yours sincerely,

Yilin Dai

Reviewer #2:

Major Comments:

  1. (line 114) “secondary mutation can overwrite the frameshift duplication as inherited” – please add relevant citations.

Response: Thank you for the comment. We have included relevant citations that support the ability of the secondary mutation that overwrites the germline frameshift duplication.

  1. In Figure 2, is there a comparison sequence from patient prior to therapy/developing resistance?

Response: Thank you for the comment. The tumor sequencing data of this patient prior to any treatment is unavailable, primarily due to the initial surgical procedure that has been carried out at an external medical facility. To address the recurrence, we performed a secondary surgical procedure followed by subsequent therapeutic interventions. All of these procedures were conducted within the premises of our institution. Nevertheless, it is noteworthy that we carried out a thorough evaluation of RAD51D expression and assessed homologous recombination level in this patient through the sectioning and staining of tumor tissues obtained from the primary surgery.

  1. Have the authors conducted invasion/migration study and expression of EMT markers to assess role of the secondary mutation and resistance? Such data may be helpful.

Response: Thank you for your insightful suggestion. Given that the primary focus of this study was to assess the impact of RAD51D mutations on patient’s response to PARP inhibitors, an extensive analysis of alterations in invasion, migration and epithelial-mesenchymal transition (EMT) markers in patients or cell lines was not extensively pursued. This topic might present a promising avenue for our future exploration.

  1. The authors can mention the duration of treatment in the captions for Figure 5.

Response: Thank you for your comment. In the revised version, we have incorporated the precise duration of drug treatment, which spans a period of 72 hours.

  1. Supplementary figure-1 should be included within the manuscript.

Response: Thank you very much for your advice. Supplementary Figure 1 has been revised and renumbered as Figure 4.

  1. Have the authors conducted western blot triplicates? If yes, the significance between the band intensity values should be plotted with relevant p-values.

Response: Thank you for your constructive advice. In the revised version, we have incorporated the replicated Western blot analysis, accompanied by the statistical analysis of band intensity values and the corresponding p-values. These results have been included in Figure 4C-D of the resubmitted manuscript.

  1. Please add a separate figure in supplementary data showing the full-length sequences with x-axis for Figure S1-c.

Response: Thank you for your suggestion. In the revised manuscript, we have included a diagram in Figure 4E illustrating the predicted alterations in the RAD51D protein sequences resulting from the primary mutation (LRG_516t1:c.270_271dup) and the secondary mutation (LRG_516t1:c.271_282del) identified in the patient.

  1. Have the authors conducted any rtPCR analysis for abnormal gene amplification in RAD51D (wt and mutation) expressing cells? Such data may be pertinent to understand the mechanism behind resistance generation.

Response: Thank you for your valuable comment. rtPCR analysis was not performed on various cell lines transduced with empty vector RAD51D-WT, RAD51D-c.270_271dup or RAD51D-c.271_282del. In the case of the frameshift mutation in RAD51D, protein expression levels were more accurately assessed using Western blot analysis. The Western blot results presented in Figure 4 clearly demonstrated the impact of RAD51D-WT, RAD51D-c.270_271dup, or RAD51D-c.271_282del on RAD51D expression. These findings indicate that the frameshift mutation in RAD51D results in the loss of RAD51D expression, while the 12bp-deletion in RAD51D restores the open reading frame of RAD51D.

  1. Please add statistical analysis section in methods. All appropriate methods/tests and software used should be noted.

Response: Thank you very much for your valuable advice. A new section on statistical analysis has been incorporated into the methods section of the resubmitted manuscript.

  1. Please list the type/model of flow cytometry used.

Response: Thank you for your valuable advice. We have included the specific type/model of flow cytometry instrument used (BD FACSAria Fusion) in the Methods section of the resubmitted manuscript.

  1. Authors need to mention where cell lines were obtained from in the methods section.

Response: Thank you for your valuable suggestion. We have explicitly stated the source of the cell lines utilized in the resubmitted manuscript. The cell lines used in this study were all purchased from the American Type Culture Collection (Manassas, VA, USA).

  1. Comments on the Quality of English Language The manuscript has a lot of typographical errors such as inclusion of unnecessary dashes in words, spelling issues etc. I would request the authors to meticulously check for all errors for corrections.

Response: Thank you for your constructive advice. We have employed professional Language Editing Services provided by MDPI to enhance the English language expression of this article.

Reviewer 3 Report

In patients with Homologous Recombination Deficiency (HRD)-positive ovarian cancer, efficacy of PARP inhibitors has been confirmed. As HR genes, mainly 15 types of genes have been identified. RAD15D is also one of the HR genes.

From December 2019 to April 2023, to explore new treatments for cancer patients, cancer genome gene panel testing (OncoGuideTM NCC Oncopanel tests: 692, FoundationOne CDx tests: 2539) were performed at national university Hospitals. Among them, 231 cases of platinum-resistant advanced serous ovarian cancer, that is, PARP inhibitor-resistant advanced serous ovarian cancer, were subjected to cancer genome gene panel testing.

As a result, 87 cases (37.7% 87/231) were HRD-positive, and 9 cases (3.9% 9/231) were found to have a pathogenic variant of RAD51D. Evidence from these clinical studies suggests that pathogenic variants in RAD51D are unlikely to be the cause of PARP inhibitor resistance.

The authors present medical information obtained from the patient's course of treatment. Authors should therefore be educated in medical ethics. Additionally, authors must state that the clinical study was approved by the institutional ethical review board. Authors must obtain informed consent from the patient.

In many experiments with cultured cells, researchers are likely to get artifactual results.

It is important for researchers to derive correct results from clinical studies using large cohorts.

In patients with Homologous Recombination Deficiency (HRD)-positive ovarian cancer, efficacy of PARP inhibitors has been confirmed. As HR genes, mainly 15 types of genes have been identified. RAD15D is also one of the HR genes.

From December 2019 to April 2023, to explore new treatments for cancer patients, cancer genome gene panel testing (OncoGuideTM NCC Oncopanel tests: 692, FoundationOne CDx tests: 2539) were performed at national university Hospitals. Among them, 231 cases of platinum-resistant advanced serous ovarian cancer, that is, PARP inhibitor-resistant advanced serous ovarian cancer, were subjected to cancer genome gene panel testing.

As a result, 87 cases (37.7% 87/231) were HRD-positive, and 9 cases (3.9% 9/231) were found to have a pathogenic variant of RAD51D. Evidence from these clinical studies suggests that pathogenic variants in RAD51D are unlikely to be the cause of PARP inhibitor resistance.

The authors present medical information obtained from the patient's course of treatment. Authors should therefore be educated in medical ethics. Additionally, authors must state that the clinical study was approved by the institutional ethical review board. Authors must obtain informed consent from the patient.

In many experiments with cultured cells, researchers are likely to get artifactual results.

It is important for researchers to derive correct results from clinical studies using large cohorts.

Author Response

Dear reviewer,

Thank you for reviewing our manuscript and providing these constructive comments. We have gone through them one by one and made revisions in the manuscript accordingly. Please see the attachment. Following is our response (in red font) to each of your comment. We have also specified the revision we made in the response. Please review and let us know your thoughts.

Best regards!

Yours sincerely,

Yilin Dai

Reviewer #3:

In patients with Homologous Recombination Deficiency (HRD)-positive ovarian cancer, efficacy of PARP inhibitors has been confirmed. As HR genes, mainly 15 types of genes have been identified. RAD15D is also one of the HR genes.

From December 2019 to April 2023, to explore new treatments for cancer patients, cancer genome gene panel testing (OncoGuideTM NCC Oncopanel tests: 692, FoundationOne CDx tests: 2539) were performed at national university Hospitals. Among them, 231 cases of platinum-resistant advanced serous ovarian cancer, that is, PARP inhibitor-resistant advanced serous ovarian cancer, were subjected to cancer genome gene panel testing.

As a result, 87 cases (37.7% 87/231) were HRD-positive, and 9 cases (3.9% 9/231) were found to have a pathogenic variant of RAD51D. Evidence from these clinical studies suggests that pathogenic variants in RAD51D are unlikely to be the cause of PARP inhibitor resistance.

The authors present medical information obtained from the patient's course of treatment. Authors should therefore be educated in medical ethics. Additionally, authors must state that the clinical study was approved by the institutional ethical review board. Authors must obtain informed consent from the patient.

In many experiments with cultured cells, researchers are likely to get artifactual results.

It is important for researchers to derive correct results from clinical studies using large cohorts.

Response: Thank you for your insightful comment. Our study was approved by the ethical committee of the Obstetrics and Gynecology Hospital of Fudan University (Number: 2021-174). Informed consent was obtained from the patient involved in the study. Written informed consent has been obtained from the patient(s) to publish this paper.

Previous studies have shown that restoration of homologous recombination (HR) repair activity might be the potential mechanisms of PARPi resistance, and highlighted the association between secondary RAD51D mutations and PARP inhibitor resistance in patients. A previous study established that the analyses of primary and secondary mutations in RAD51C and RAD51D offer supports for primary mutations in conferring sensitivity to PARPi, while secondary mutations serve as a mechanism for acquiring resistance to PARPi (Cancer Discov. 2017 Sep;7(9):984-998. doi: 10.1158/2159-8290.CD-17-0419). Our discovery aligns with this result, as we found that secondary mutations in RAD51D, which result in the reestablishment of open reading frames and homologous recombination function, could indeed play a role in conferring PARP inhibitor resistance in our case.

In order to improve the quality of English language, we have employed professional Language Editing Services provided by MDPI to enhance the English language expression of this article.

Round 2

Reviewer 2 Report

The authors have satisfactorily answered most questions raised by reviewers and have made appropriate corrections to the manuscript. No further comments on the state of the revised manuscript.